# AF$_2$: ADAPTIVE FOCUS FRAMEWORK FOR AERIAL IMAGERY SEGMENTATION

## ABSTRACT

As a specific semantic segmentation task, aerial imagery segmentation has been widely employed in high spatial resolution (HSR) remote sensing images understanding. Besides common issues (e.g. large scale variation) faced by general semantic segmentation tasks, aerial imagery segmentation has some unique challenges, the most critical one among which lies in foreground-background imbalance. There have been some recent efforts that attempt to address this issue by proposing sophisticated neural network architectures, since they can be used to extract informative multi-scale feature representations and increase the discrimination of object boundaries. Nevertheless, many of them merely utilize those multi-scale representations in ad-hoc measures but disregard the fact that the semantic meaning of objects with various sizes could be better identified via receptive fields of diverse ranges. In this paper, we propose Adaptive Focus Framework (AF$_2$), which adopts a hierarchical segmentation procedure and focuses on adaptively utilizing multi-scale representations generated by widely adopted neural network architectures. Particularly, a learnable module, called Adaptive Confidence Mechanism (ACM), is proposed to determine which scale of representation should be used for the segmentation of different objects. Comprehensive experiments show that AF$_2$ has significantly improved the accuracy on three widely used aerial benchmarks, as fast as the mainstream method.

## 1 INTRODUCTION

Understanding geospatial objects, such as plants, buildings, vehicles, etc., in high spatial resolution (HSR) remote sensing images plays a vital role in land cover monitoring, urban management, and civil engineering. Aerial imagery segmentation, as a specific semantic segmentation task that assigns a semantic category to each image pixel, has been widely leveraged in HSR remote sensing images understanding since it can provide semantic and location information for objects of interest.

Nonetheless, in addition to some common issues in most semantic segmentation datasets (Caesar et al., 2018; Cordts et al., 2016; Zhou et al., 2019), including large scale variation (Kirillov et al., 2019; Long et al., 2015; Ronneberger et al., 2015), complex scene (Chen et al., 2017; Zhao et al., 2017), and indistinguishable object boundaries (Cheng et al., 2020; Kirillov et al., 2020; Zhen et al., 2020), aerial imagery segmentation has its own challenges, the most critical one among which lies in foreground-background imbalance (Deng et al., 2019; 2018; Li et al., 2021; Pang et al., 2019; Waqas Zamir et al., 2019; Xia et al., 2018; Zheng et al., 2020). Taking images in Fig. 1 as examples, the foreground proportion can be extremely small, e.g. less than 1% for the leftmost image. Such acute imbalance could drastically increase the difficulty of object recognition, as even human eyes can hardly recognize them from the image. Moreover, the larger intra-class variance of the background objects may significantly increase the risk of false positive results (Li et al., 2021; Zheng et al., 2020).

Existing general semantic segmentation methods mainly pay attention to designing sophisticated neural network architectures that can obtain informative multi-scale feature representations (Chen et al., 2018; He et al., 2016; Kirillov et al., 2019; Sun et al., 2019; Szegedy et al., 2017) and highlight the object boundaries (Kirillov et al., 2020; Li et al., 2021; Zhen et al., 2020). To further address the foreground-background imbalance challenge of aerial imagery segmentation, some recent efforts have created more delicate modules in the neural networks to obtain superior results. For instance, Foreground-Aware Relation Network (FarSeg) (Zheng et al., 2020) introduces a foreground-scene

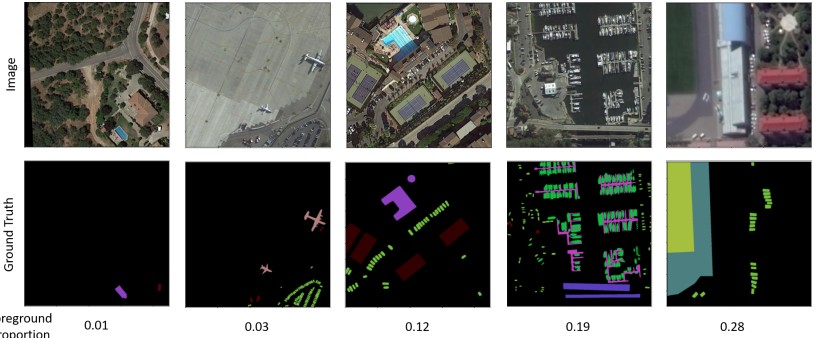

Figure 1: Illustration of aerial imagery segmentation.

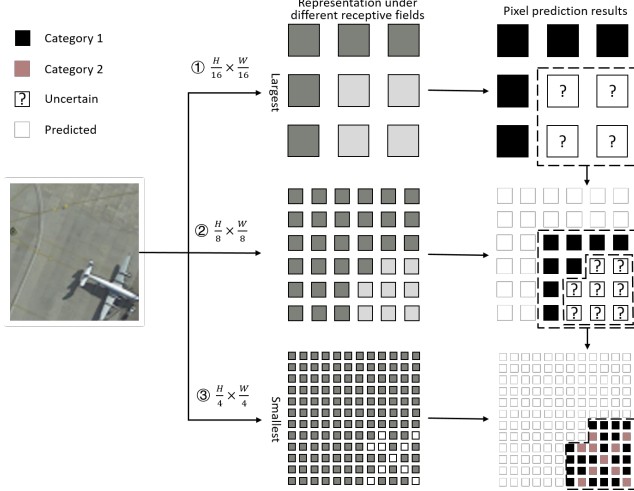

Figure 2: Diagrammatic representation of Adaptive Focus Framework.

relation module to enhance the discrimination of foreground features as well as a foreground-aware optimization to alleviate foreground-background imbalance problem. More recently, PointFlow (Li et al., 2021) designs a dual point matcher to select points from the salient area and object boundaries.

While these previous studies have demonstrated the effectiveness of sophisticated neural network architecture design in both general and aerial imagery segmentation, most of them ignore the other side of the coin, i.e. how to efficiently utilize the multi-scale representations generated by complex neural networks. Intuitively, to identify the semantic meaning of a large object, it is more important for the model to leverage the representations obtained by wider receptive fields since they can represent the semantics of the whole large object rather than confusing the discrimination with interior details. On the other hand, for small objects, it is more efficient for the model to exploit the representations obtained by more concentrated receptive fields because they can focus on the discrimination without being distracted by noisy context. Unfortunately, most of the existing general or aerial segmentation methods merely employ multi-scale representations in ad-hoc ways, either simply concatenating them or arbitrarily using the final layer. This inevitably limits the potential of aerial imagery segmentation.

Inspired by the adaptively-focusing process of the human eye (Artal et al., 2006; Wikipedia contributors, 2021), we propose a novel Adaptive Focus Framework (AF$_2$) in this paper, which adopts a hierarchical segmentation procedure for aerial imagery segmentation. The general idea of this framework is shown in Fig. 2. Particularly, through any widely employed model structure (e.g. encoder-decoder structure), AF$_2$ can obtain hierarchical representation maps based on different levels of receptive fields. The semantic segmentation procedure starts from the representation map created by the largest receptive field. After obtaining the segmentation result on this level, AF$_2$ will filter out the pixels of low confidence on segmentation and then carry out another round of segmentation

procedure on a finer-grained feature map. The whole process will be conducted repeatedly until all the pixels have been predicted, or until there is no finer-grained feature map. In $AF_2$, an Adaptive Confidence Mechanism (ACM) is proposed to deal with pixel filtration. The confidence of a pixel is defined as the highest value among the probabilities that the pixel belongs to each category, while an adaptive updated threshold is set to filter out low confidence pixels.

In summary, the main contributions of this paper include:

- We propose Adaptive Focus Framework ($AF_2$), which adopts a hierarchical segmentation procedure and focuses on adaptively utilizing multi-scale representations generated by widely adopted neural network architectures.
- In the pixel filtering process for each scale's representation, the Adaptive Confidence Mechanism (ACM) is adopted to dynamically decide which pixels need to use finer-grained features. This mechanism ensures the performance and robustness of the framework.
- Extensive experiments and analysis demonstrate the advantage of $AF_2$. It has significantly improved the accuracy on three typical aerial benchmarks. At the same time, its convergence speed and inference speed are also the fastest.

It is worth noting that $AF_2$ is a general framework, focusing particularly on efficient representation utilization, which can be leveraged to benefit any other semantic segmentation task.

## 2 RELATED WORKS

### 2.1 GENERAL SEMANTIC SEGMENTATION

Fully-convolutional networks (FCNs Long et al. (2015)) are the earliest method of using deep learning to model semantic segmentation problem. The backbone models (He et al., 2016; Simonyan & Zisserman, 2014; Sun et al., 2019; Szegedy et al., 2017; 2016; Xie et al., 2017) are used to generate a lower resolution output than the input image and use bilinear up-sampling to recover the original image resolution. Employing the dilated convolution (Chen et al., 2017; 2018; Yu & Koltun, 2015) to replace the down-sampling operation will have a better performance at the expense of more memory and computation cost. The spatial context information can overcome the limited receptive field of convolution layer to a certain extent such as Atrous Spatial Pyramid Pooling (ASPP Chen et al. (2017; 2018)), Pyramid Pooling Module (PPM Zhao et al. (2017)) , Densely connected Atrous Spatial Pyramid Pooling (DenseASPP Yang et al. (2018)) , Relation-Augmented fully convolutional network (RA-Net Mou et al. (2019)), etc. PointRend (Kirillov et al., 2020) performs point-based segmentation predictions at adaptively selected locations based on an iterative subdivision algorithm. Some other works (Li et al., 2020a; Yuan et al., 2020; Zhang et al., 2020) propose architectures specific for segmentation boundary that is difficult to predict.

The encoder-decoder architectures (Chen et al., 2018; Kirillov et al., 2019; Lin et al., 2017a; Ronneberger et al., 2015; Takikawa et al., 2019; Li et al., 2019) progressively upsample the high-level features and combine them with the features from lower levels, ultimately generating high-resolution features. For instance, Deeplab v3+ (Chen et al., 2018) combines dilated convolutions with an encoder-decoder structure to produce the output on a grid 4x sparser than the input. SemanticFPN (Kirillov et al., 2019) merges the information from all levels of the Feature Pyramid Network (FPN) pyramid into a single output and produces a dense prediction.

### 2.2 SEMANTIC SEGMENTATION FOR AERIAL IMAGERY

In recent years, employing deep learning to accelerate the understanding of aerial image has received widespread attention (Kaiser et al., 2017; Kussul et al., 2017; Marcos et al., 2018; Marmanis et al., 2018; Scott et al., 2017), and benefits a lot of applications such as agriculture vision (Arakeri et al., 2016; Kamilaris & Prenafeta-Boldú, 2018; Patrício & Rieder, 2018), road extraction (Bastani et al., 2018; Ji et al., 2018), land cover mapping (Li et al., 2016; Malkin et al., 2018; Robinson et al., 2019), forest monitor (Jiao et al., 2019; Yuan et al., 2015), etc. They often design delicate structures to ensure that general semantic segmentation migrates well for specific application scenarios. For instance, Relation Augmented network (RA-Net Mou et al. (2019)) proposes a spatial relation module and a channel relation module to explicitly model global relations. Foreground-Aware Relation Network

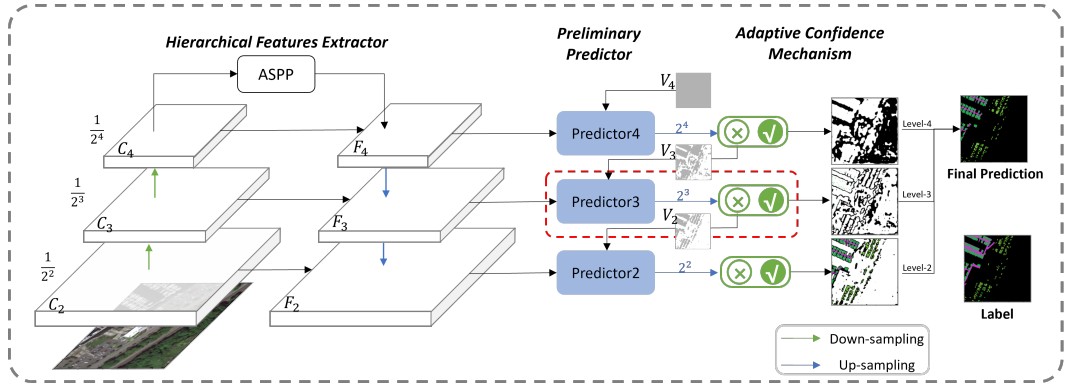

Figure 3: Adaptive Focus Framework. $C_l$ and $F_l$ represent different hierarchy feature maps. $V_l$ means the pixel set fed into level $l$ and we show them in gray. The adaptive confidence mechanism (ACM) is employed to judge whether the prediction for each pixel is sufficiently confident or not in each level, and ✓ is the sign of confidence. The detail of the red dashed box is shown in Fig. 4.

(FarSeg Zheng et al. (2020)) enhances the discrimination of foreground features and proposes a balanced optimization based on focal loss (Lin et al., 2017b). PointFlow (Li et al., 2021) designs a dual point matcher to select points from the salient area and object boundaries.

## 3 METHOD

In this section, we discuss details of the proposed AF$_2$. As shown in Fig. 3, AF$_2$ consists of three main parts: hierarchical features extractor, predictor, and adaptive confidence mechanism (ACM). The pseudo-code of Adaptive Focus Framework is shown in appendix.

- **Hierarchical features extractor:** Given an input image, this is used to extract the hierarchical feature maps of the image. The higher the level of feature map, the coarser the granularity of feature map and the larger the receptive field.

- **Preliminary predictor:** For each level of the feature map, there is a preliminary predictor to output the categories of the pixels that still cannot be confidently predicted with higher-level feature maps.

- **Adaptive confidence mechanism for prediction selection:** The adaptive confidence mechanism (ACM) is employed to judge whether the prediction for each pixel is sufficiently confident or not in each level. The predictions with high enough confidence will be accepted for the corresponding pixels. Otherwise, the pixels will be passed down to the next hierarchy for further prediction with finer-grained features, until there is no pixels for prediction or the lowest level is reached.

### 3.1 HIERARCHICAL FEATURES EXTRACTOR

Hierarchical feature extractor is the fundamental part of the AF$_2$. It prepares different levels of feature maps for further pixel-level classification. The higher the level of feature map, the larger the receptive field. Since AF$_2$ is independent of the particular structure of the network, any mainstream feature extraction network can be adopted as long as it is capable of extracting a hierarchy of feature maps.

The typical model structures, such as Fully-Convolutional Network (FCN Long et al. (2015)), Feature Pyramid Networks (FPN Lin et al. (2017a)) , Semantic FPN Kirillov et al. (2019), etc., can be employed here. In this section, we take FPN (Lin et al., 2017a) with Atrous Spatial Pyramid Pooling module (ASPP) (Chen et al., 2017) as an example. Specifically, ResNet (He et al., 2016) is chosen as the backbone network for basic feature extraction. We denote the different level of feature maps generated by ResNet as $\mathcal{C} = \left\{ C_l \big| C_l \in \mathbb{R}^{d_l \times \frac{H}{2^l} \times \frac{W}{2^l}}, l \in [L_{\min}, L_{\max}] \right\}$, where $H$ and $W$ represent the image's original height and width. $l$ is the level of feature map, while $L_{\min}$ and $L_{\max}$ are the lowest and highest levels (e.g. if the output stride of ResNet is $2^5$, its $L_{\max}$

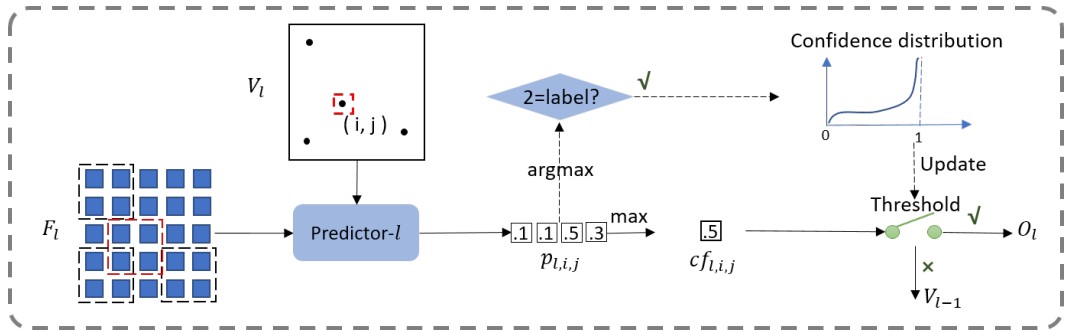

Figure 4: The process of Adaptive Confidence Mechanism. Firstly, the corresponding feature map of pixels in $V_l$ are fed into predictor $l$ to get predicted probabilities. Then, the threshold is employed to judge whether the pixel should go down the hierarchy to get lower-grained features or output the prediction. Meanwhile, the threshold is periodically updated according to the confidence distribution of correctly predicted pixels. Note that the threshold will be fixed without update during the inference.

is 5. The lowest feature map's output stride is always $2^2$, and its $L_{min}$ is 2.). Furthermore, let $\mathcal{F} = \left\{ F_l \middle| F_l \in \mathbb{R}^{d_l \times \frac{H}{2^l} \times \frac{W}{2^l}}, l \in [L_{\min}, L_{\max}] \right\}$, which stands for the the feature map set from the decoder part. $F_l$ is defined as

$$F_l = \begin{cases} f\left( \text{ASPP}(C_l), C_l \mid \theta_l \right), & l = L_{\max} \\ f\left( \text{UP}_2(F_{l+1}), C_l \mid \theta_l \right), & \text{otherwise.} \end{cases} \tag{1}$$

where $f(\cdot, \cdot \mid \theta_l)$ represents the top-down feature fusion function. ASPP represents Atrous Spatial Pyramid Pooling Module (Chen et al., 2017). $\text{UP}_2$ represents the bilinear up-sampling function with a scale factor 2.

## 3.2 PRELIMINARY PREDICTOR

Multiple pixel-level preliminary predictors are assigned for the different feature maps, and the goal of the predictor is to output the category of the pixels that cannot be confidently predicted with higher-level features. Specifically, the feature map $F_l$ is fed into its corresponding predictor for level $l$ to obtain the prediction results. Since the results belong to $\mathbb{R}^{n \times \frac{H}{2^l} \times \frac{W}{2^l}}$ ($n$ is the category number), the bilinear up-sampling method is employed to generate a prediction with the same size as the original image. The whole process can be formulated as[1]:

$$P_l = \text{SOFTMAX}\left( \text{UP}_{2^l}\left( g\left( F_l \mid \theta_l \right) \right) \right), \tag{2}$$

where $g(\cdot \mid \theta_l)$ is the prediction function based on multi-layer perceptron (MLP), and $\text{UP}_{2^l}$ is the bilinear up-sampling method with a scale factor $2^l$.

In fact, only a part of the prediction results will be selected from $P_l$. Specifically, we denote $p_{l,i,j}$ as the prediction results vector in $P_l$ for pixel $(i, j) \in V_l$, where $V_l$ is the pixel set in the original image that still cannot be confidently predicted in higher levels like Eq. equation 4. Only these selected prediction results, i.e. $\{p_{l,i,j} \mid (i, j) \in V_l\}$, will be processed by ACM to determine whether they are adopted as the final results.

## 3.3 ADAPTIVE CONFIDENCE MECHANISM FOR PREDICTION SELECTION

The purpose of adaptive confidence mechanism (ACM) is to determine which pixels can be identified and which pixels need to be fed into the lower level for prediction. To achieve the above, the metric for each pixel is defined and is named as the pixel's confidence. Then, the filtration function is adopted based on this metric. The process of ACM is depicted in Fig. 4.

Firstly, the pixel's confidence is defined to evaluate whether the corresponding prediction result is reliable enough (i.e. the larger the pixel's confidence, the more reliable the prediction). For the

---

[1]Certainly, $P_l = \text{SOFTMAX}\left( f\left( \text{UP}_{2^l}\left( F_l \right) \mid \theta \right) \right)$ is another option which performs up-sampling first.

pixel $(i, j) \in V_l$, the prediction result $p_{l,i,j}$ is an $n$-dimensional vector to indicate the probabilities belonging to the $n$ categories. We define the highest value among these probabilities as its confidence[2], that is $cf_{l,i,j} = \max (p_{l,i,j}), (i, j) \in V_l$.

Secondly, we filter out some of the pixels to be fed into the lower level according to their confidence. A straightforward way is to select the pixels with the top $k$-lowest confidence in each image. However, since the variance among different images, especially for aerial images, is very large, the top-$k$ approach cannot efficiently handle such complicated situation. For instance, pixels in complicated images usually have low confidence, while pixels in simple images have relatively higher confidence. Therefore, we choose to use a uniform threshold for all images. Since the accuracy of prediction is improved during training, the threshold should also be learnable and adaptively updated with model training. More concretely, the statistical information of the confidence for correctly predicted pixels in each level is employed to help update the threshold adaptively. That is

$$\tau_l^{t+1} = \gamma \cdot \tau_l^t + (1 - \gamma) \cdot \text{QUANTILE}_r \left\{ cf_{l,i,j} \mid (i, j) \in V_l \wedge \arg\max(p_{l,i,j}) = label_{i,j} \right\}, \quad (3)$$

where $\tau_l^t$ is the threshold in step $t$ for level $l$, $\gamma$ is soft update factor, and $\text{QUANTILE}_r$ represents the $r$-quantile method. It should be noted that the threshold is fixed without update in inference.

Thus, compared with threshold $\tau_l$, the pixels with a lower confidence will be fed into the level $l - 1$ like Eq. equation 4 except for the lowest level. Meanwhile, the categories of the rest pixels will be determined based on the prediction results of this layer like Eq. equation 5. In the lowest level $L_{min}$, we set $\tau_{L_{min}} = 0$ so that all the pixels $V_{L_{min}}$ will be determined.

$$V_{l-1} = \{(i, j) \mid (i, j) \in V_l \wedge cf_{l,i,j} < \tau_l^t\}, l \neq L_{min}, \quad (4)$$

$$O_l = \{(i, j, \arg\max(p_{l,i,j})) \mid (i, j) \in V_l \wedge cf_{l,i,j} \geq \tau_l^t\}, \quad (5)$$

where $O_l$ is the confident prediction in each level.

Since there are a number of pixels whose categories could be determined in each level, i.e., $O_l$ for $l_{min} \leq l \leq l_{max}$, the final prediction results is a union of all these pixels $O = \bigcup_{l=L_{min}}^{L_{max}} O_l$.

## 3.4 OPTIMIZATION

The optimization objective function is based on cross-entropy loss. Due to the hierarchical prediction procedure adopted in $AF_2$, the overall loss is accumulated at each level of the hierarchical predictions. Since pixels in $V_l$ are fed into predictor in level $l$, the total loss function can be formulated as follow:

$$J = \sum_{l \in [L_{min}, L_{max}]} \left( \frac{1}{||V_l||} \sum_{(i,j) \in V_l} \text{CROSS-ENTROPY} (p_{l,i,j}, label_{i,j}) \right). \quad (6)$$

## 4 EXPERIMENTS

To evaluate the proposed method, we carry out comprehensive experiments on three aerial imagery datasets: iSAID, Vaihingen and Potsdam. We first introduce these datasets. Then, we show the experimental results on those four datasets. After that, we conduct further analysis to examine the importance of each components of $AF_2$. The implementation details are shown in appendix.

## 4.1 DATASETS

**iSAID.** iSAID (Waqas Zamir et al., 2019; Xia et al., 2018) is the largest dataset for instance segmentation in the HSR remote sensing imagery. It contains 2,806 high-resolution aerial images with 655,451 instance annotations from 15 categories. iSAID is distinguished from other semantic segmentation datasets for its significant imbalance between the annotated instances and background as well as its scale variation even for the instances from the same category. In our experiments, we follow its default dataset split where 1,411 images are used for training, 458 for validation, and 937 for testing. The original images can be as large as 4000×13000 pixels. Following previous work (Zheng et al.,

---

[2]The probability distribution entropy, the difference value between the first and second largest probability value, etc. are other optional choices for pixel's confidence.

Table 1: Experiments on 3 aerial imagery datasets. For iSAID $val$ set, we show the mIoU score and the category with a significant improvement, such as: baseball court (BC), large vehicle (LV), helicopter (HC), swimming pool (SP) and roundabout (RA). All the experiments use ResNet-50 with weights pretrained on ImageNet as backbone for fair comparison except HRNet (Sun et al., 2019).

| Method | iSAID (%) | | | | | | Vaihingen (%) | | Potsdam (%) | |
|---|---|---|---|---|---|---|---|---|---|---|
| | mIoU | BC | LV | HC | SP | RA | mIoU | m-$F_1$ | mIoU | m-$F_1$ |
| FCN (Long et al., 2015) | - | - | - | - | - | - | 64.2 | 75.9 | 73.1 | 83.1 |
| PSPNet (Zhao et al., 2017) | 60.3 | 61.1 | 58.0 | 10.9 | 46.8 | 68.6 | 65.1 | 76.8 | 73.9 | 83.9 |
| Ocnet (Yuan et al., 2018) | - | - | - | - | - | - | 65.7 | 77.4 | 74.2 | 84.1 |
| DenseASPP (Yang et al., 2018) | 57.3 | 54.8 | 55.6 | 33.4 | 37.5 | 53.4 | 64.7 | 76.4 | 73.9 | 83.9 |
| Deeplabv3+ (Chen et al., 2018) | 61.5 | 56.6 | 60.3 | 34.5 | 41.4 | 65.1 | 64.3 | 76.0 | 74.1 | 83.9 |
| SemanticFPN (Kirillov et al., 2019) | 62.1 | 54.1 | 61.0 | 37.4 | 42.8 | 70.2 | 66.3 | 77.6 | 74.3 | 84.0 |
| RefineNet (Cheng et al., 2020) | 60.2 | 61.1 | 58.2 | 23.0 | 43.4 | 65.6 | - | - | - | - |
| UPerNet (Xiao et al., 2018) | 63.8 | 55.3 | 61.3 | 30.3 | 45.7 | 68.7 | 66.9 | 78.7 | 74.3 | 84.0 |
| HRNet (Sun et al., 2019) | 61.5 | 59.4 | 62.1 | 14.9 | 44.2 | 52.9 | 66.9 | 78.2 | 73.4 | 83.4 |
| GSCNN (Takikawa et al., 2019) | 63.4 | 56.1 | 63.8 | 33.8 | 48.8 | 58.5 | 67.7 | 79.5 | 73.4 | 84.1 |
| SFNet (Li et al., 2020b) | 64.3 | 58.8 | 62.9 | 30.4 | 47.8 | 69.8 | 67.6 | 78.6 | 74.3 | 84.0 |
| RANet (Mou et al., 2019) | 62.1 | 53.2 | 60.1 | 38.1 | 41.8 | 70.5 | 66.1 | 78.2 | 73.8 | 83.9 |
| PointRend (Kirillov et al., 2020) | 62.8 | 55.4 | 62.3 | 29.8 | 45.0 | 66.0 | 65.9 | 78.1 | 72.0 | 82.7 |
| FarSeg (Zheng et al., 2020) | 63.7 | 62.1 | 60.6 | 35.8 | 51.2 | 71.4 | 65.7 | 78.0 | 73.4 | 83.3 |
| PointFlow (Li et al., 2021) | 66.9 | 62.2 | 64.6 | 37.9 | 50.1 | 71.7 | 70.4 | 81.9 | **75.4** | **84.8** |
| AF$_2$-AFPN | **67.8** | **66.2** | **67.3** | **38.9** | **53.1** | **77.0** | **70.5** | **82.1** | 74.9 | 84.4 |

2020; Li et al., 2021), these images are cropped into patches with a fixed size of 896×896 with a sliding window striding 512 pixels, and these models are trained with 16 epochs on cropped images for all experiments. We employ the mean intersection over union (mIoU) as evaluation metric.

**Vaihingen and Potsdam.** Vaihingen includes 33 aerial images with 2494×2064 pixels. Potsdam includes 38 aerial images with 6000×6000 pixels. 6 categories are defined for both of them. Following the previous work (Li et al., 2021), images are cropped into patches with fixed sizes of 768×768 and 896×896, respectively. These models are trained with 200 epochs for all experiments. We use mIoU and mean-$F_1$ metrics to evaluate the proposed method.

## 4.2 RESULT COMPARISON

In this section, we conduct detailed experiments on iSAID, Vaihingen and Potsdam datasets to compare the final prediction performance and the inference efficiency of AF$_2$ with several mainstream methods. The base model for hierarchical feature extractor adopted is a combination of FPN (Lin et al., 2017a) and ASPP (Chen et al., 2018), abbreviated as AFPN for convenience. Our whole implementation is denoted as AF$_2$-AFPN.

**Result Comparison on iSAID.** As shown in Table. 1, AF$_2$-AFPN achieves the state-of-the-art result among all previous works with $r = 0.32$ in ACM. The results also show a significant improvement among some categories, which demonstrates again that AF$_2$ can identify the tiny instance with high accuracy. Meanwhile, compared with previous methods, It shows that AF$_2$-AFPN has exceeded most mainstream models in terms of running efficiency. More details can be found in the appendix.

It is worth noting that the architecture re-design methods, such as FarSeg (Zheng et al., 2020) and the best model before PointFlow (Li et al., 2021), can be easily integrated with our model to further improve the performance. However, tweaking the architecture is not the focus of this work. Meanwhile, the source code of some models, e.g. PointFlow, is not available yet. As one piece of future work, we will verify the effectiveness of the combination of AF$_2$ and other more advanced architectures.

**Result Comparison on Vaihingen and Potsdam.** To further verify the performance of our framework, we also conduct experiments on two well-known aerial imagery datasets, i.e. Vaihingen and Potsdam. Compared with iSAID dataset, the categories are relatively balanced in terms of instance shape and category ratio in these two datasets. The quantitative results listed in Table 1 show that AF$_2$ outperforms all methods except PointFlow. More details can be found in the appendix.

| Image | Ground Truth | Level-4 | Level-3 | Level-2 | Final Prediction (Stacked) |
|---|---|---|---|---|---|

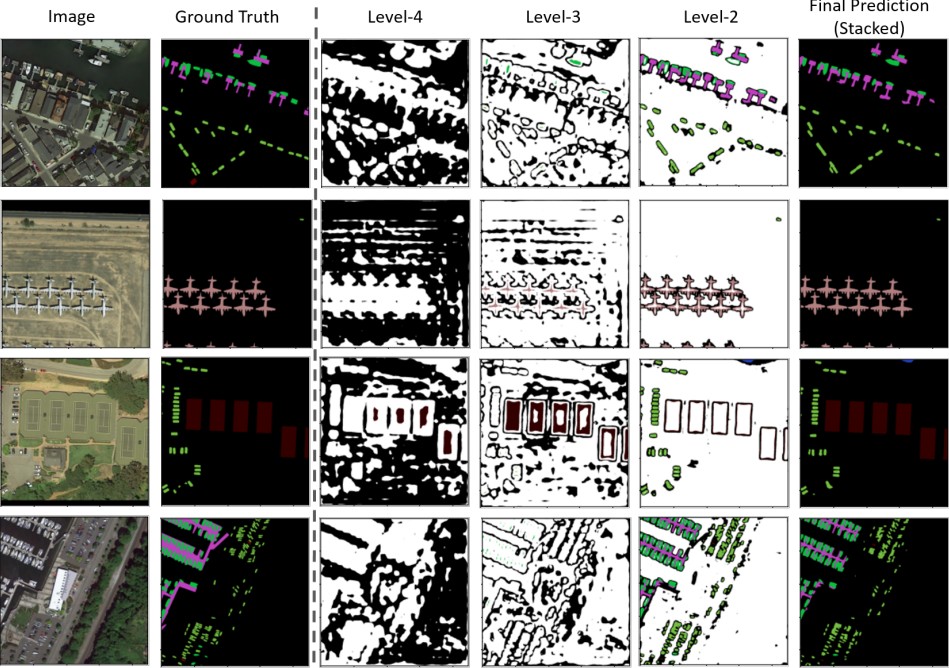

Figure 5: Image prediction process show case on iSAID validation set with $r = 0.32$. Different color means different category. Black means the background. Specifically, white means that the pixel has not been predicted or has been predicted in higher level.

### 4.3 SENSITIVITY ANALYSIS

In this section, we conduct thorough analysis over iSAID to study the importance of each modules of $AF_2$. To better demonstrate the analysis results, we first present the process of image prediction over some examples from the validation set of iSAID in Figure 5. In this figure, columns 3 to 5 indicate the accepted prediction results at level 4 to 2, respectively, while the last column is the stacked final results. As we can see, at higher levels, it is more often the inner areas of the background or foreground that are predicted successfully. Lower level finer-grained features are primarily used to improve the prediction results of the edges. This hierarchical prediction method is independent to the proportion of foreground and background and thus can effectively solve the problems existed in the previous method.

**Study on the Effect of the ACM.** ACM, as a core part of $AF_2$, is designed to filter out the low confidence pixels. In ACM, the quantile ratio is a key factor to control the filtration threshold. To study its influence, we set different values of $r$ in ACM. As shown in Table 2, the models integrated with $AF_2$ achieve significant improvement over the basic AFPN model. In addition, the experimental results show that the setting of $r$'s value has a great influence on the results. Particularly, either too large or too small $r$ will skew the model toward finer- or coarser- grained representations. Experimental results show that 0.3 is a compromise value on the iSAID dataset, which can greatly improves the prediction accuracy over the baseline method. It is worth noting that, in the case of $r = 0.9$, where most pixels (96.0%) are segmented in level-2, the mIoU value has been improved by 2.1%. This result indicates that the loss function for coarse-grained features is beneficial.

**Study on the Effect of Employed Levels.** In this experiment, we study the influence on performance when employing feature maps from only a subset of levels in $AF_2$. By default, feature maps from level-2, level-3, and level-4 are employed in the framework to produce the final result. However, as shown in Table3, when using only two levels' feature maps, the segmentation mIoU dropped from 67.4 to 66.8 even in the best case. Note that only employing feature maps from level-2 and level-4 is equivalent to FPN and FCN respectively. Experimental results demonstrate that the multi-scale feature maps utilization is crucial.

Table 2: Study of ACM. The prediction ratio means how many pixels have completed prediction at this level (i.e. $\frac{\|O_l\|}{H \times W}$). The foreground ratio means foreground pixels proportion of the current pixel sets (i.e. $\frac{\|\{label_{i,j} \ is \ fg. \ | \ (i,j) \in V_l\}\|}{\|V_l\|}$).

| Method | $r$ | Prediction Ratio (%) | | | Foreground Ratio (%) | | | mIoU (%) |
|---|---|---|---|---|---|---|---|---|
| | | level-4 | level-3 | level-2 | level-4 | level-3 | level-2 | |
| AFPN | - | - | - | 100 | - | - | 3.3 | 63.3 |
| AF$_2$-AFPN | 0.9 | 2.2 | 1.8 | 96.0 | 3.3 | 3.5 | 3.6 | 65.4 |
| AF$_2$-AFPN | 0.7 | 9.7 | 16.3 | 74.0 | 3.3 | 3.8 | 4.3 | 65.8 |
| AF$_2$-AFPN | 0.5 | 32.7 | 41.2 | 26.0 | 3.3 | 4.7 | 8.8 | 66.4 |
| AF$_2$-AFPN | 0.3 | 73.7 | 20.0 | 6.3 | 3.3 | 8.8 | 27.6 | **67.4** |
| AF$_2$-AFPN | 0.1 | 91.4 | 7.0 | 1.6 | 3.3 | 22.7 | 41.3 | 64.7 |

**Hierarchical Features Extractor with Different Neural Architectures.** In this experiment, we aim to evaluate the extendibility and flexibility of the proposed AF$_2$ in terms of feature extraction. Specifically, we select 3 typical architectures: FCN (Long et al., 2015), SemanticFPN (Kirillov et al., 2019), FPN+ASPP (Chen et al., 2017; Lin et al., 2017a) as the hierarchical features extractor, which are named as FCN, SFPN and AFPN for convenience. For FCN, we combine the feature of each level with all of its higher level feature maps to introduce richer semantic information into lower levels. As shown in Table 4, we find that the different architectures have different degrees of score improvement with the assistance of AF$_2$. AF$_2$ narrows the gap among different architectures. Even for the the FCN architecture, which only has a bottom-up feature generation, AF$_2$ can achieve about 8.4 improvement on mIoU score. This result further illustrates the extendibility and superiority of AF$_2$.

Table 3: Study of Employed Levels.

| Method | Level-4 | Level-3 | Level-2 | mIoU |
|---|---|---|---|---|
| AF$_2$-AFPN | ✓ | | | 57.9 |
| AF$_2$-AFPN | | ✓ | | 60.7 |
| AF$_2$-AFPN | | | ✓ | 63.1 |
| AF$_2$-AFPN | ✓ | ✓ | | 66.0 |
| AF$_2$-AFPN | | ✓ | ✓ | 66.7 |
| AF$_2$-AFPN | ✓ | | ✓ | 66.8 |
| AF$_2$-AFPN | ✓ | ✓ | ✓ | **67.4** |

Table 4: Study of Hierarchical Features Extractor.

| Method | mIoU |
|---|---|
| FCN | 57.9 |
| SFPN | 62.1 |
| AFPN | 63.3 |
| AF$_2$-FCN | 66.3 |
| AF$_2$-SFPN | 66.8 |
| AF$_2$-AFPN | **67.4** |

**Results on general segmentation benchmark** We further verify our approach on general segmentation dataset Cityscapes. As show in Table 5, the experiment shows that our method has about 2% mIoU improvement. The training, validation, testing data is 2975, 500, and 1525 respectively. We only use the fine-data for training. During training, data augmentation contains random horizontal flip, random cropping with the size 768×768. We train a totally 50k iterations with a minibatch 16. We use ResNet-50 with weights pretrained on ImageNet.

Table 5: Experiment on Cityscapes validation.

| Methods | mIoU | road | SW | BD | wall | fence | pole | TL | TS | VG | terrain | sky | person | rider | car | truck | bus | train | MT | bicycle |
|---|---|---|---|---|---|---|---|---|---|---|---|---|---|---|---|---|---|---|---|---|
| SFPN | 0.76 | 0.98 | 0.85 | 0.92 | 0.52 | 0.61 | 0.63 | 0.69 | 0.78 | 0.92 | 0.64 | 0.95 | 0.82 | 0.63 | 0.95 | 0.71 | 0.82 | 0.58 | 0.68 | 0.77 |
| AF2-SFPN | 0.78 | 0.98 | 0.86 | 0.93 | 0.57 | 0.62 | 0.66 | 0.71 | 0.80 | 0.93 | 0.65 | 0.95 | 0.83 | 0.66 | 0.95 | 0.76 | 0.84 | 0.70 | 0.67 | 0.78 |

SW, BD, TL, TS, VG, MT represents sidewalk, building, traffic light, traffic sign, vegetation, and motorcycle, respectively.

## 5 CONCLUSION

In this paper, we argue that the lack of efficient utilization of multi-scale representations could be a bottleneck for accurate semantic segmentation on HSR aerial imagery, which is characterized by the huge scale variation of objects and the imbalance between foreground and background. We present AF$_2$, i.e. Adaptive Focus Framework, to alleviate this critical but long-standing concern. AF$_2$ is independent of the specific architecture and is capable of adaptively utilizing multi-scale feature representations and producing the final result through the proposed Adaptive Confidence Mechanism. Extensive experiments and analyses have demonstrated its remarkable advantages in boosting segmentation accuracy and have proved its universality on common architectures and datasets.

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

# A APPENDIX

## A.1 PSEUDO-CODE OF ADAPTIVE FOCUS FRAMEWORK

Pseudo-code is shown in Alg. 22 which summarizes and describes how various stages work in training or testing mode.

---
**Algorithm 1** Pseudo-code of Adaptive Focus Framework

---
**Input:** Image: $X \in R^{c \times H \times W}$
**Output:** Image segmentation
1: Initialize: $V_{L_{max}} = \{(i,j) \mid i \in [1,H], j \in [1,W]\}$
2: Mode: train or test
3: Feature map from FPN: $\{F_l \mid F_l \in \mathbb{R}^{d_l \times \frac{H}{2^l} \times \frac{W}{2^l}}, l \in [L_{\min}, L_{\max}]\}$
4: Preliminary predictor result: $P_l = Softmax\left(Up_{2^l}\left(g\left(F_l \mid \theta_l\right)\right)\right)$
5: **for** $l \in \{L_{max}, L_{max-1}, ..., L_{min}\}$ **do**:
6:     **for** each $(i,j) \in V_l$ **do**:
7:         select pixel (i,j) 's preliminary prediction from $P_l$: $p_{l,i,j} = P_{l,(i,j)}$
8:         calculate the confidence value: $cf_{l,i,j} = \max\left(p_{l,i,j}\right), (i,j) \in V_l$
9:         **if** $cf_{l,i,j} > \tau_l$ and $l \neq L_{min}$ **then**:
10:            put $(i, j, \arg\max(p_{l,i,j}))$ into $O_l$
11:         **else**
12:            put $(i, j)$ into $V_{l-1}$
13:         **end if**
14:         **if** mode is train **then**:
15:            Adaptively update the threshold:$\tau_l$
16:         **end if**
17:     **end for**
18: **end for**
19: Final prediction is: $O = \bigcup_{l=L_{\min}}^{L_{\max}} O_l$
20: **if** mode is train **then**:
21:     optimize objective function: $J = \sum_l \frac{1}{||V_l||} \sum_{(i,j) \in V_l} CrossEntropy(p_{l,i,j}, label_{i,j})$
22: **end if**

---

## A.2 IMPLEMENTATION DETAILS

Following Chen et al. (2018); Li et al. (2021); Mou et al. (2019); Zheng et al. (2020), we use ResNet-50 with weights pretrained on ImageNet in all the experiments for fair comparison except for HRNet (Sun et al., 2019). The output stride of the backbone is adjusted to 16 by setting the convolution stride of the last layer to 1 and employing dilated convolution following DeepLab v3+(Chen et al., 2018). The feature maps from the backbone, i.e. $C_2$, $C_3$, $C_4$, are the outputs of the $conv2\_x$, $conv3\_x$, and $conv5\_x$ of the ResNet-50 respectively. All the confidence thresholds are initialized to be 0.5. In the threshold update, we set the soft intensity $\gamma$ to 0.9 and $r$ to 0.3 for QUANTILE$_r$. The learning rate decreases from 0.01 to 0.0001 following the poly policy, i.e. $lr_{step} = lr_{init}(1 - \frac{step}{max\_step})^{power}$, where $power$ is 0.9. We employ synchronized SGD over 4 GPUs with each mini-batch containing 16 cropped patches, weight decay of 0.0001 and momentum of 0.9. The synchronized batch normalization is enabled for cross-GPU communication. Following Li et al. (2021); Mou et al. (2019); Zheng et al. (2020), for iSAID, we adopt data augmentation during the training, which includes horizontal flip, vertical flip, and rotations of 90, 180, and 270 degrees. Note that, for each experimental setting, we run the experiment 5 times and report the average score to remove the influence of randomness.

## A.3 RESULTS ON ISAID DATASET

We show the detailed results on iSAID Dataset in Tab. 6, and AF$_2$ achieves state-of-the-art for most categories.

Table 6: Experimental results on iSAID val set. The bold values in each column represent the best entries. The category are defined as: ship (Ship), storage tank (ST), baseball diamond (BD), tennis court (TC),baseball court (BC), ground field track (GTF), bridge (Bridge), large vehicle (LV), small vehicle (SV), helicopter (HC), swimming pool (SP), roundabout (RA), soccerball field (SBF), plane (Plane), harbor (Harbor). All the models are trained under the same setting following the FarSeg(Zheng et al., 2020) and PointFlow (Li et al., 2021).

| Method | backbone | mIoU(%) | Ship | ST | BD | TC | BC | GTF | Bridge | LV | SV | HC | SP | RA | SBF | Plane | Harbor |
|---|---|---|---|---|---|---|---|---|---|---|---|---|---|---|---|---|---|
| PSPNet (Zhao et al., 2017) | ResNet50 | 60.3 | 65.2 | 52.1 | 75.7 | 85.6 | 61.1 | 60.2 | 32.5 | 58.0 | 43.0 | 10.9 | 46.8 | 68.6 | 71.9 | 79.5 | 54.3 |
| DenseASPP (Yang et al., 2018) | RenNet50 | 57.3 | 55.7 | 63.5 | 67.2 | 81.7 | 54.8 | 52.6 | 34.7 | 55.6 | 36.3 | 33.4 | 37.5 | 53.4 | 73.3 | 74.7 | 46.7 |
| Deeplabv3+ (Chen et al., 2018) | ResNet50 | 61.5 | 63.2 | 67.8 | 69.9 | 85.3 | 56.6 | 52.9 | 34.2 | 60.3 | 43.2 | 34.5 | 41.4 | 65.1 | 73.8 | 81.0 | 52.3 |
| SemanticFPN (Kirillov et al., 2019) | ResNet50 | 62.1 | 68.9 | 62.0 | 72.1 | 85.4 | 54.1 | 48.9 | 44.9 | 61.0 | 48.6 | 37.4 | 42.8 | 70.2 | 61.6 | 81.7 | 54.9 |
| RefineNet (Cheng et al., 2020) | ResNet50 | 60.2 | 63.8 | 58.6 | 72.3 | 85.3 | 61.1 | 52.8 | 32.6 | 58.2 | 42.4 | 23.0 | 43.4 | 65.6 | 74.4 | 79.9 | 51.1 |
| UPerNet (Xiao et al., 2018) | ResNet50 | 63.8 | 68.7 | 71.0 | 73.1 | 85.5 | 55.3 | 57.3 | 43.0 | 61.3 | 45.6 | 30.3 | 45.7 | 68.7 | 75.1 | 84.3 | 56.2 |
| HRNet (Sun et al., 2019) | HRNetW18 | 61.5 | 65.9 | 68.9 | 74.0 | 86.9 | 59.4 | 61.5 | 33.8 | 62.1 | 46.9 | 14.9 | 44.2 | 52.9 | **75.6** | 81.7 | 52.2 |
| GSCNN (Takikawa et al., 2019) | ResNet50 | 63.4 | 65.9 | 71.2 | 72.6 | 85.5 | 56.1 | 58.4 | 40.7 | 63.8 | 51.1 | 33.8 | 48.8 | 58.5 | 72.5 | 83.6 | 54.4 |
| SFNet (Li et al., 2020b) | ResNet50 | 64.3 | 68.8 | 71.3 | 72.1 | 85.6 | 58.8 | 60.9 | 43.1 | 62.9 | 47.7 | 30.4 | 47.8 | 69.8 | 75.1 | 83.1 | 57.3 |
| RANet (Mou et al., 2019) | ResNet50 | 62.1 | 67.1 | 61.3 | 72.5 | 85.1 | 53.2 | 47.1 | **45.3** | 60.1 | 49.3 | 38.1 | 41.8 | 70.5 | 58.8 | 83.1 | 55.6 |
| PointRend (Kirillov et al., 2020) | ResNet50 | 62.8 | 64.4 | 69.9 | 73.7 | 82.9 | 55.4 | **61.1** | 38.5 | 62.3 | 48.1 | 29.8 | 45.0 | 66.0 | 72.7 | 80.7 | 54.0 |
| FarSeg (Zheng et al., 2020) | ResNet50 | 63.7 | 65.4 | 61.8 | 77.7 | 86.4 | 62.1 | 56.7 | 36.7 | 60.6 | 46.3 | 35.8 | 51.2 | 71.4 | 72.5 | 82.0 | 53.9 |
| PointFlow (Li et al., 2021) | ResNet50 | 66.9 | 70.3 | 74.7 | 77.8 | 87.7 | 62.2 | 59.5 | 45.2 | 64.6 | 50.2 | 37.9 | 50.1 | 71.7 | 75.4 | 85.0 | 59.3 |
| AF$_2$-AFPN | ResNet50 | **67.8** | 69.5 | 73.7 | 80.9 | 89.9 | 66.2 | 56.5 | 41.1 | 67.3 | 52.0 | 38.9 | 53.1 | 77.9 | 74.4 | **84.5** | **59.6** |

**Efficiency Comparison:** we compare the model size and inference speed on validation set in Figure 6. It shows that AF$_2$-AFPN has exceeded most mainstream models in terms of running efficiency.

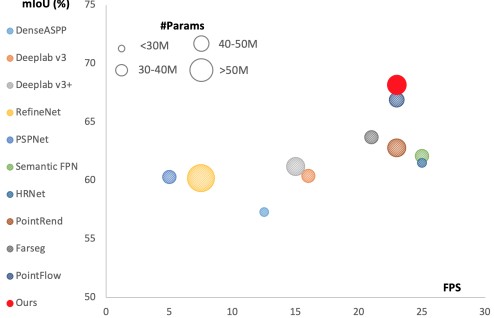

Figure 6: Speed (FPS) versus accuracy (mIoU) on iSAID *val* set.

## A.4 RESULTS ON VAIHINGEN DATASET

Vaihingen contains 33 images (of different sizes) and 6 categories have been defined. Following previous work (Li et al., 2021), we adopt large patches as the iSAID dataset. We utilize 16 images for training and the rest 17 images for testing. For training set, the image IDs are 1, 3, 5, 7, 11, 13, 15, 17, 21, 23, 26, 28, 30, 32, 34, 37. For validation set, the images IDs are 2, 4, 6, 8, 10, 12, 14, 16, 20, 22, 24, 27, 29, 31, 33, 35, 38. We crop the images into 768×768 with a sliding window striding 512 pixels, and all the experiments are trained with 200 epochs. Like previous work, we use the mIoU and m-$F_1$ (i.e. the harmonic mean of precision and recall) as the main metric.

The detailed results on Vaihingen Dataset are shown in Tab. 7. The process of prediction is shown in Fig. 7. In Vaihingen dataset, the imbalance between foreground and background is relatively slight. Among all categories, only the cars are tiny, while the background is complex. Our framework also has a big improvement in these categories, which demonstrates the flexibility of AF$_2$.

## A.5 RESULTS ON POTSDAM DATASET

Potsdam contains 38 images (of different sizes) and 6 categories have been defined. Following (Li et al., 2021), we utilize 24 images for training, and the image IDs are 2_10, 2_11, 2_12, 3_10, 3_11, 3_12, 4_10, 4_11, 4_12, 5_10, 5_11, 5_12, 6_7, 6_8, 6_9, 6_10, 6_11, 6_12, 7_7, 7_8, 7_9, 7_10, 7_11, 7_12. We utilize the another 14 images for test, and the image IDs are 2_13, 2_14, 3_13, 3_14,

Table 7: Experimental results on the Vaihingen Dataset. The results are reported with single scale input. The bold values in each column represent the best entries. The category are defined as: impervious surfaces (Imp.surf.), buildings (Build), low vegetation (Low veg), trees (Tree), cars (Car), cluster/background (Cluster). All the models are trained under the same setting following the PointFlow (Li et al., 2021).

| Method | mIoU(%) | mean-$F_1$ | $F_1$ per category | | | | | |
|---|---|---|---|---|---|---|---|---|
| | | | Imp.surf. | Build. | Low veg. | Tree | Car | Cluster |
| FCN (Long et al., 2015) | 64.2 | 75.9 | 87.6 | 91.6 | 77.8 | 84.6 | 73.5 | 40.3 |
| PSPNet (Zhao et al., 2017) | 65.1 | 76.8 | 88.4 | 92.8 | 79.2 | 85.9 | 73.5 | 41.0 |
| OCNet(ASP-OC) (Yuan et al., 2018) | 65.7 | 77.4 | 88.8 | 92.9 | 79.2 | 85.8 | 73.9 | 43.8 |
| Denseaspp (Yang et al., 2018) | 64.7 | 76.4 | 87.3 | 91.1 | 76.2 | 83.4 | 77.1 | 43.3 |
| Deeplabv3+ (Chen et al., 2018) | 64.3 | 76.0 | 88.7 | 92.8 | 78.9 | 85.6 | 72.4 | 37.6 |
| SemanticFPN (Kirillov et al., 2019) | 66.3 | 77.6 | 89.6 | 93.6 | 79.7 | 86.3 | 75.7 | 40.7 |
| UPerNet (Xiao et al., 2018) | 66.9 | 78.7 | 89.2 | 93.0 | 79.4 | 86.0 | 74.9 | 49.7 |
| HRNet-W18 (Sun et al., 2019) | 66.9 | 78.2 | 89.2 | 92.6 | 78.7 | 85.7 | 77.1 | 45.9 |
| GSCNN (Takikawa et al., 2019) | 67.7 | 79.5 | 89.4 | 92.6 | 78.8 | 85.4 | 77.9 | 52.9 |
| SFNet (Li et al., 2020b) | 67.6 | 78.6 | 90.0 | **94.0** | 80.3 | **86.5** | 78.9 | 41.9 |
| RANet (Mou et al., 2019) | 66.1 | 78.2 | 88.0 | 92.3 | 79.1 | 86.0 | 78.8 | 53.1 |
| PointRend (Kirillov et al., 2020) | 65.9 | 78.1 | 88.2 | 92.4 | 78.9 | 84.5 | 73.5 | 51.1 |
| FarSeg (Zheng et al., 2020) | 65.7 | 78.0 | 88.0 | 92.0 | 78.2 | 85.2 | 73.3 | 51.5 |
| PointFlow (Li et al., 2021) | 70.4 | 81.9 | **90.1** | 93.6 | 77.7 | 85.4 | **80.0** | **64.6** |
| AF$_2$-AFPN | **70.5** | **82.1** | 89.6 | 93.4 | **79.8** | 86.2 | 79.8 | 63.5 |

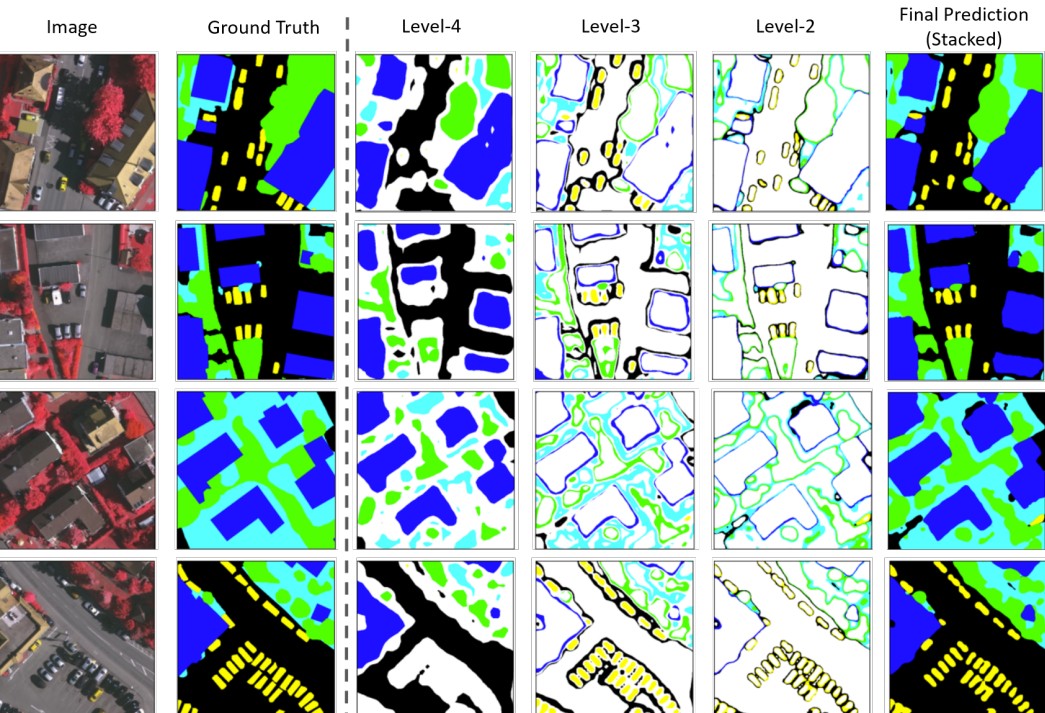

Figure 7: Image Prediction Process Show Case on Vaihingen *val* set.

Table 8: Experimental results on the Potsdam Dataset. The results are reported with single scale input. The bold values in each column represent the best entries. The category are defined as: impervious surfaces (Imp.surf.), buildings (Build), low vegetation (Low veg), trees (Tree), cars (Car), cluster/background (Cluster). All the models are trained under the same setting following the PointFlow (Li et al., 2021).

| Method | mIoU(%) | mean-$F_1$ | $F_1$ per category | | | | | |
|---|---|---|---|---|---|---|---|---|
| | | | Imp.surf. | Build. | Low veg. | Tree | Car | Cluster |
| FCN (Long et al., 2015) | 73.1 | 83.1 | 90.2 | 94.7 | 84.1 | 85.6 | 89.2 | 54.8 |
| PSPNet (Yang et al., 2018) | 73.9 | 83.9 | 90.8 | 95.4 | 84.5 | 86.1 | 88.6 | 58.0 |
| OCnet(ASP-OC) (Yuan et al., 2018) | 74.2 | 84.1 | 90.9 | 95.5 | 84.8 | 86.0 | 89.2 | 58.2 |
| Denseaspp (Yang et al., 2018) | 73.9 | 83.9 | 90.8 | 95.4 | 84.6 | 86.0 | 88.5 | 58.1 |
| Deeplabv3+ (Chen et al., 2018) | 74.1 | 83.9 | 91.0 | 95.6 | 84.6 | 86.0 | 90.0 | 56.2 |
| SemanticFPN (Kirillov et al., 2019) | 74.3 | 84.0 | 91.0 | 95.5 | 84.9 | 85.9 | 90.4 | 56.3 |
| UPerNet (Xiao et al., 2018) | 74.3 | 84.0 | 90.9 | 95.7 | 85.0 | 86.0 | 90.2 | 56.2 |
| HRNet-W18 (Sun et al., 2019) | 73.4 | 83.4 | 90.4 | 94.9 | 84.2 | 85.4 | 90.0 | 55.5 |
| GSCNN (Takikawa et al., 2019) | 73.4 | 84.1 | 91.4 | 95.5 | 84.8 | 85.8 | 91.2 | 55.9 |
| SFNet (Li et al., 2020b) | 74.3 | 84.0 | 91.0 | 95.5 | 85.1 | 86.0 | 90.9 | 55.5 |
| RANet (Mou et al., 2019) | 73.8 | 83.9 | 90.8 | 92.1 | 84.3 | 86.8 | 88.9 | 56.0 |
| PointRend (Kirillov et al., 2020) | 72.0 | 82.7 | 89.8 | 94.6 | 82.8 | 85.2 | 85.2 | 58.6 |
| FarSeg (Zheng et al., 2020) | 73.4 | 83.3 | 90.7 | 95.2 | 84.3 | 85.3 | 90.2 | 54.1 |
| PointFlow (Li et al., 2021) | 75.4 | 84.8 | 91.5 | 95.9 | 85.4 | 86.3 | 91.1 | 58.6 |
| AF$_2$-AFPN | 74.9 | 84.4 | 91.1 | 95.7 | 85.0 | 86.2 | 91.5 | 57.0 |

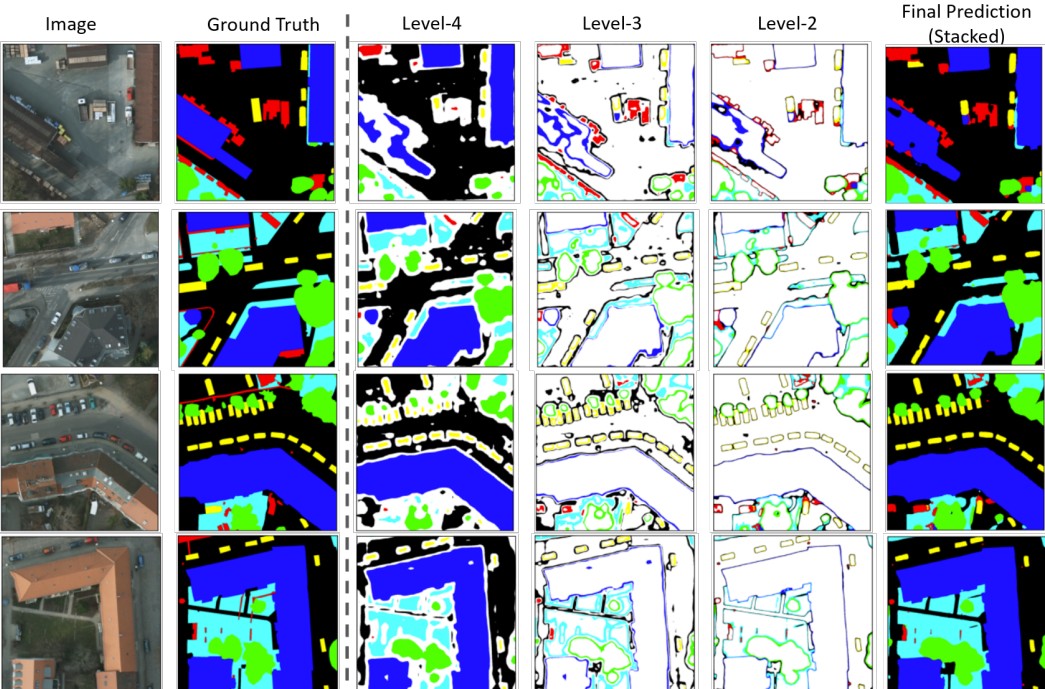

Figure 8: Image Prediction Process Show Case on Potsdam $val$ set.

4_13, 4_14, 4_15, 5_13, 5_14, 5_15, 6_13, 6_14, 6_15, 7_13. We crop the images into $896 \times 896$ with a sliding window striding 512 pixels, and all the experiments are trained with 80 epochs. Like previous work, we use the mIoU and m-$F_1$ as the main metric.

We show detailed results in Tab. 8 and the process of prediction in Fig. 8 for Potsdam dataset.

