# OpenReview forum: "AF$_2$: Adaptive Focus Framework for Aerial Imagery Segmentation"
_ICLR.cc/2022/Conference — ICLR 2022 Submitted_

### Official Review · Reviewer_464N · 2021-10-27

**Correctness:** 3
**Technical Novelty And Significance:** 2
**Empirical Novelty And Significance:** 2
**Recommendation:** 5
**Confidence:** 3

**Main Review:**


The strength of the approach is that experimental results show improvement against state of  the art for mIoU score.
Intuitively a multi-resolution approach (pyramid) to segment objects of various size is sound.

However the paper lacks clarity and coherence:
- foreground/background are mentioned extensively in the paper but these are not defined. How are these relating to the classes defined with the dataset used in the experiments?
- class imbalance (between foreground/ background )   is mentioned as an issue in the abstract but it is unclear how this method addresses class imbalance.
- the abstract also indicates that many techniques "disregard the fact that the semantic meaning of objects with various sizes could be better identified via receptive fields of diverse ranges" but it is unclear how the presented work addresses this shortcoming:  how semantic meaning of objects and association with size is used in the proposed pipeline? What are the objects segmented ? how are their real sizes translating into number of  pixels in the dataset used (  in aerial images are geolocated and correspond to an area (e.g. pixel == 50cm x 50cm) ) and relation to receptive fields of filters used?   In the results reported what objects correspond to "small object" and "large object" (e.g. a few objects are mentioned in Tab 1 , and as well in Tab.5)?
- Several sentences lack clarity e.g.  "For instance, pixels in complicated images usually have low confidence, while pixels in simple images have relatively higher confidence." What is a simple/complicated image?
- mathematics are difficult to follow e.g. Eq 3


**Summary Of The Paper:**

This paper proposes a pipeline for image segmentation. The  approach is validated  experimentally for remote sensing applications.
A hierarchical segmentation procedure corresponding to a Feature Pyramid Networks (FPN) with Atrous  Spatial Pyramid is used.
 A preliminary detector take advantage of these feature maps for each resolution level (=layer) for computing class prediction to each pixel. In addition a confidence measurement to each pixel level prediction is also computed (Adaptive confidence mechanism).
The architecture is optimized using the cross entropy.
Results show that the proposed pipeline improves the mIoU  scores compared to the state of the art techniques on three standard dataset.


**Summary Of The Review:**


The proposed method looks promising in improving segmentation results in remote sensing application.
However clarity has to be improved.

---

### Official Review · Reviewer_wDBh · 2021-10-30

**Correctness:** 4
**Technical Novelty And Significance:** 2
**Empirical Novelty And Significance:** 2
**Recommendation:** 5
**Confidence:** 5

**Main Review:**

Strong Points:

1,  The paper is easy to follow and understanding.
2,  The proposed methods achieve the state-of-the-art results on two ariel datasets(iSAID, Vaihingen).


Weakness:
Sorted by the importance.  Here are the details:

1, Several related works are not cited and compared.  These work may decrease the novelty of this paper.
The difference of this work and these dynamic network only lies on the prediction source.
This work adpot the predictions as route gating which are used in [1][2][3][4].

[1] Learning Dynamic Routing for Semantic Segmentation CVPR-2020
[2] Fine-Grained Dynamic Head for Object Detection NIPS-2020
[3] Dynamic Routing Networks arxiv-2019
[4] Improving Video Instance Segmentation via Temporal Pyramid Routing arxiv-2021

Using predictions for further refine is not very novel and may bring extra computation cost.


2, Several statements are not true.
"e.g. PointFlow, is not available yet"
I search for the Github. https://github.com/lxtGH/PFSegNets This work is already opensourced about half year ago.


3, Several statements are not necessary which may be decribed in short sentences.

Sec.3.1 and Sec.3.4 may be not necessary since these are the common knowledges for segmentation.

4,  Several detailed design are missing. The choices of ASPP should be claimed.
Why use ASPP in the head?
What about the GFlops and FPS of proposed methods?
What is the baseline in Tab3?

5, What about the generalization of Adaptive Confidence Mechanism? Will it still work well on other scene understanding tasks or datasets?
If not, it may be not suitable for the ICLR submission.



**Summary Of The Paper:**

This paper propose Adaptive Focus Framework (AF2), which adopts a hierarchical segmentation procedure and focuses on adaptively utilizing multiscale representations generated by widely adopted neural network architectures.
The main contribution is a learnable module, called Adaptive Confidence Mechanism (ACM) to determine which scale of representation should be used for the segmentation of different objects. The method achieves the state-of-the-art results on two ariel datasets(iSAID, Vaihingen).

**Summary Of The Review:**

Compared with exsiting methods on scene understanding, the proposed Adaptive Confidence Mechanism is little weak.

---

### Official Review · Reviewer_hDsQ · 2021-11-01

**Correctness:** 3
**Technical Novelty And Significance:** 2
**Empirical Novelty And Significance:** 2
**Recommendation:** 3
**Confidence:** 4

**Main Review:**

1. To my understanding, this paper essentially proposes a new strategy to merge multi-scale predictions, by choosing the high-confidence output of earlier levels, and passing low-confidence output to downstream levels. Similar strategy could be seen as early as Viola-Jones cascade classifier, but in the domain of semantic segmentation, I do not remember seeing similar design, so in that sense this paper does have some small novelty, though it's incremental.

2. That said, the ideas of merging multi-scale predictions in semantic segmentation have already been studied in many existing work, from the early FCN [1], to more focused studies on this such as [2] and [3]. The proposed AF2 can be considered a hand-crafted strategy to decide which output scale to take, which could be better than naive method like average-pooling, but I'm not convinced it will work better than learnable attention such as in [2].

   2.1. Even comparing to max-pooling, it's possible that AF2 will settle on one class in early scale after passing the threshold, but later scale may reveal an even higher confidence from a different class.

   2.2. The main problem of the paper is it failed to compare to any of these multi-scale merging strategy. For example, [2] claimed the attention method can boost baseline model result by 6%, which is higher than AF2's boost on AFPN (these are not really comparable due to different settings, but I meant the paper should have a fair comparison on this aspect).

   - [1] Fully Convolutional Networks for Semantic Segmentation, CVPR 2015

   - [2] Attention to Scale: Scale-aware Semantic Image Segmentation, CVPR 2016

   - [3] Hierarchical Multi-Scale Attention for Semantic Segmentation, 2020

3. Besides, a few other issues or questions of the paper:

   3.1. The paper highlights the "adaptive" threshold and claims top-k won't be effective on aerial images, but there is no experiment designed specifically to validate how much improvements the adaptive mechanism can bring over baselines like fixed threshold or top-k.

   3.2. How is the "soft update factor" in Eq.(3) decided?

   3.3. Is r set to a fixed value for experiments on all datasets?

   3.4. In Sec. 4.3, the paper states "This hierarchical prediction method is independent to the proportion of foreground and background and thus can effectively solve the problems existed in the previous method." - I don't understand the logic here. I agree the hierarchical design is independent to the foreground/background ratio, but then how can it also help solving imbalance issue if they're independent?

**Summary Of The Paper:**

This paper proposes a hierarchical segmentation procedure for aerial imagery semantic segmentation. The proposed method adaptively filters pixels in each scale representation by dynamically adjusting the filtering confidence to decide pixels will be segmented by the next level. Experiments have shown the proposed method can boost existing model architectures' performance to achieve SotA on several datasets.

**Summary Of The Review:**

The paper proposed a method to selectively merge multi-scale predictions, but failed to compare to any other methods aiming for similar goal (e.g. max-pooling, attention). There is no experiment to validate the "adaptive" design of the mechanism either (compared to fixed threshold or top-k). Therefore, I recommend reject.

---

### Official Review · Reviewer_VRMT · 2021-11-02

**Correctness:** 4
**Technical Novelty And Significance:** 1
**Empirical Novelty And Significance:** 1
**Recommendation:** 3
**Confidence:** 4

**Details Of Ethics Concerns:**

No ethics concerns

**Main Review:**


This paper proposes  Adaptive Focus Framework (AF2) to deal with semantic segmentation task of aerial imagery. The strengths of this paper is that the writing is clear and easy to understand. The structure of this paper is ok. The weakness is that the idea is simple and not novel enough. In addition, the experiments cannot verify the effectiveness of the proposed method.

**Summary Of The Paper:**

In this paper, the Adaptive Focus Framework (AF2), which adopts a hierarchical segmentation procedure and focuses on adaptively utilizing multiscale representations generated by widely adopted neural network architectures, is proposed. Particularly, a learnable module, called Adaptive Confidence Mechanism (ACM), is proposed to determine which scale of representation should be used for the segmentation of different objects.

**Summary Of The Review:**

The proposed AF2 tries to solve HSR aerial imagery semantic segmentation by making use of multi-scale representations. The whole  Adaptive Focus Framework (AF2) is simple, including Hierarchical features extractor, Preliminary predictor and Adaptive confidence mechanism for prediction selection (ACM).
As the first two modules are popular in semantic segmentation and other related works, the main contribution of this work is the proposed ACM, which judges whether the prediction for each pixel is sufficiently confident or not in each level.  But actually, the proposed ACM is simple lacking novelty. The experiments in this work cannot verify the effectiveness of proposed method solidly. In Table 1, AF2-AFPN brings slightly better performance compared with PointFlow. On Potsdam, its performance is even worse. For general segmentation benchmark, only SFPN method is reported and compared, but in PointFlow, it has mIoU 80.3% on Cityscape.

All in all, i think this paper is not novel and solid enough to be accepted by ICLR.

---

### Decision · Program_Chairs · 2022-01-20

**Decision:**

Reject

**Comment:**

Four experts reviewed this paper and all recommended rejection. There was no rebuttal. The reviewers raised many concerns regarding the paper, such as missed citations, lack of comparison with related methods, and some presentation issues. Considering the reviewers' concerns, we regret that the paper cannot be recommended for acceptance at this time.  The authors are encouraged to consider the reviewers' comments when revising the paper for submission elsewhere.